# Electrospun Composite Nanofibrous Materials Based on (Poly)-Phenol-Polysaccharide Formulations for Potential Wound Treatment

**DOI:** 10.3390/ma13112631

**Published:** 2020-06-09

**Authors:** Lidija Fras Zemljič, Uroš Maver, Tjaša Kraševac Glaser, Urban Bren, Maša Knez Hrnčič, Gabrijela Petek, Zdenka Peršin

**Affiliations:** 1Laboratory for Characterization and Processing of Polymers, Faculty of Mechanical Engineering, University of Maribor, Smetanova 17, SI-2000 Maribor, Slovenia; tjasha.sternad@gmail.com (T.K.G.); zdenka.persin@gmail.com (Z.P.); 2Faculty of Medicine, Institute of Biomedical Sciences and Department of Pharmacology, University of Maribor, Taborska ulica 8, SI-2000 Maribor, Slovenia; uros.maver@um.si; 3Faculty of Chemistry and Chemical Engineering, University of Maribor, Smetanova 17, SI-2000 Maribor, Slovenia; urban.bren@um.si (U.B.); masa.knez@um.si (M.K.H.); 4Faculty of Electrical Engineering and Computer Science, University of Maribor, Smetanova 17, SI-2000 Maribor, Slovenia; gabrijela.petek@um.si; 5The BISTRA Scientific Research Centre Ptuj, Slovenski trg 6, SI-2250 Ptuj, Slovenia

**Keywords:** electrospinning, chitosan, catechin, resveratrol, antioxidant, antimicrobial, wound healing

## Abstract

In this paper, we focus on the preparation of electrospun composite nanofibrous materials based on (poly)-phenol-polysaccharide formulation. The prepared composite nanofibres are ideally suited as a controlled drug delivery system, especially for local treatment of different wounds, owing to their high surface and volume porosity and small fibre diameter. To evaluate the formulations, catechin and resveratrol were used as antioxidants. Both substances were embedded into chitosan particles, and further subjected to electrospinning. Formulations were characterized by determination of the particle size, encapsulation efficiency, as well as antioxidant and antimicrobial properties. The electrospinning process was optimised through fine-tuning of the electrospinning solution and the electrospinning parameters. Scanning electron microscopy was used to evaluate the (nano)fibrous structure, while the successful incorporation of bio substances was assessed by X-ray Photoelectron Spectroscopy and Fourier transform infrared spectroscopy. The bioactive properties of the formed nanofibre -mats were evaluated by measuring the antioxidative efficiency and antimicrobial properties, followed by in vitro substance release tests. The prepared materials are bioactive, have antimicrobial and antioxidative properties and at the same time allow the release of the incorporated substances, which assures a promising use in medical applications, especially in wound care.

## 1. Introduction

Wounds have been called “the silent epidemic”, as their annual incidence in the EU27 affects around 4 million patients. In a typical hospital today, between 25% and 50% of beds are occupied by patients with wounds, up to 60% of which are non-healing wounds such as infected surgical wounds, pressure sores and leg/foot ulcers [1,2].

Healthcare providers have faced and continue to face major problems in managing wounds and wound related problems for centuries, despite the extensive knowledge available. Chronic wounds are strongly correlated with age, diseases and surgical procedures, reflecting the increasing need for innovative treatments [2,3].

An ideal wound dressing should be biocompatible and biodegradable, promote the healing process, be able to swell and absorb excess exudate and have a high porosity that allows breathing. Dressings for today’s chronic wounds come in many forms [2,3,4,5,6,7], and ultrafine and nanofibres have the potential to revolutionise wound care. Nanofibres are produced by various methods, but among them, electrospinning is an environmentally friendly technology that requires very simple equipment and permits the use of a variety of polymers, both synthetic and natural, resulting in different functionalities of the final fibre products [2,8,9,10,11]. In order to promote wound healing, electrospun nanofibre mats (as described in research and projects) can be produced in various ways [2,12,13,14,15,16,17,18,19,20,21,22,23,24,25,26,27].

The healing process of infected wounds is a very complicated process in which wounds are colonised by bacteria that form a biofilm, usually resulting in aggressive wound management, which is complicated by a very limited range of therapeutic interventions (either drugs or other means). Biofilms are particularly difficult to treat with antibiotics because they have numerous defence mechanisms. In addition, antibiotic-resistant bacteria are a growing problem in wound infections. Research is therefore being carried out on natural biopolymers and polyphenols. In this context, curcumin was incorporated into the polycaprolactone (PCL) [28] as well as cellulose acetate (CA) [29] nanofibres which support the attachment and proliferation of human skin fibroblasts and show a high rate of wound closure. Nanofibre mats with incorporated plant components, with emphasis on essential oils, honey [10,30,31] and herbal extracts from olive leaves [18,23,32] were developed and showed efficient antioxidant and antimicrobial activity against the most common bacteria found in infected wounds.

Among biopolymers, chitosan is very attractive for electrospinning [33] and shows promising potential for use in tissue engineering and regenerative medicine [34], while among other biopolymers it has little or no antioxidant capacity, which is essential for wound healing. Natural compounds and extracts have been identified as antioxidants that help to inhibit free radicals and oxidative damage in tissues and food. A large number of papers on antioxidants charged on electrospun nanofibres have been categorised and reviewed to identify applications and new trends. The active compounds used repeatedly were tannic acid (polyphenol), quercetin (flavonoid), curcumin (polyphenol) and vitamin B6 (pyridoxine) [10]. The incorporation of active compounds into nanofibres often improves their bioavailability and gives them increased stability, changes the mechanical properties of polymers, increases the biocompatibility of nanofibres and introduces bioactivity [10].

Although the number of research groups working on the development of new electrospun materials for tissue engineering and wound dressings is increasing rapidly, it is still an ambitious task to effectively combine different natural antimicrobial agents and antioxidants in novel electrospun nanomats [2,10,35]. The aim of this study was to achieve synergistic behaviour of chitosan and (poly)-phenols to reduce a broad spectrum of pathogenic bacteria while inhibiting free radicals present in wounds. To this end, catechin and resveratrol, which are known to be the strongest protectors against symptoms of ageing and free radical damage [36,37,38], were incorporated into chitosan nanoparticles by the ionic gelation method. Nanoparticle-based formulations have been characterised by the determination of particle size (dynamic light scattering DLS), zeta potential, encapsulation efficiency and the evaluation of antioxidant and antimicrobial efficacy. Samples of medical composite textiles were prepared by electrospinning from the liquid formulations produced. The spinning solution and electrospinning parameters were optimised under this aspect. The prepared samples were further characterised by scanning electron microscopy (SEM), X-ray photoelectron spectroscopy (XPS) and attenuated total reflectance Fourier transform infrared spectroscopy (ATR-FTIR), followed by the evaluation of the antioxidants and the antimicrobial effectiveness using standard techniques.

## 2. Materials and Methods

### 2.1. Materials’ Preparation

#### 2.1.1. Liquid Formulations

The following liquid solutions were prepared, as listed below.

Chitosan solution (CS)

Low-molecular weight chitosan (Mw = 82,000), deacetylated (75–85%) chitin, Poly (D-glucosamine) from Sigma-Aldrich (Vienna, Austria); was used. A chitosan solution was prepared with a weight concentration of 10 g/L. An appropriate amount of chitosan (powder) was weighed and suspended in deionised water. To dissolve chitosan, the pH of the solution was adjusted to 3.8 with the addition of concentrated acetic acid. The prepared solution was stirred on a magnetic stirrer for 24 h at room temperature, and then adjusted to pH 3.5 with concentrated acetic acid.

Catechin solution (CAT)

For the preparation of the catechin solution, (+) catechin hydrate was used from Sigma Aldrich (PhytoLab GmbH & Co. KG, Vestenbergsgreuth, Germany). A solution of catechin was prepared with a weight concentration of 10 g/L. An appropriate amount of (+) catechin hydrate (powder) was weighed and dissolved in 100% ethanol. The prepared solution was stirred for 20 min on a magnetic stirrer.

Resveratrol solution (RES)

Resveratrol (3,4,5-trihydroxy-trans-stilbene) was used from Sigma-Aldrich (PhytoLab GmbH & Co. KG, Vestenbergsgreuth, Germany). A solution of resveratrol was prepared with a weight concentration of 20 g/L. The appropriate amount of resveratrol (powder) was weighed and dissolved in 100% ethanol. The prepared solution was stirred for 20 min on a magnetic stirrer.

The concentrations of both antioxidant solutions were prepared in accordance with their solubility.

Dispersion of chitosan nanoparticles (CSNP)

Chitosan nanoparticles were prepared by the ionic gelation technique. Simultaneously, 0.2% (*w*/*v*) of sodium tripolyphosphate (TPP) solution was added to a fixed volume of 1% (*w*/*v*) chitosan solution, in order to obtain a 5:1 chitosan to TPP weight ratio. This ratio was chosen according to the previously published work, which reported it as an optimal ratio for obtaining the desirable antimicrobial activity of nanoparticles’ dispersion [39].

Preparation of (poly)-phenol-loaded (catechin/resveratrol) chitosan–TPP nano dispersion

Chitosan nanoparticles were prepared by the ionic gelation technique as mentioned above. Simultaneously, 0.2% (*w*/*v*) of sodium tripolyphosphate (TPP) solution and (poly)-phenols (i.e., 10 mg/mL of catechin (CAT) or 20 mg/mL resveratrol (RES) solution) were added to a fixed volume of 1% (*w*/*v*) chitosan solution. Particles were formed spontaneously under magnetic stirring for 1 h at room temperature. The final pH of CS nanoparticles’ dispersions with embedded (poly)-phenols (i.e., with incorporated catechin (CSNP_CAT) and with incorporated resveratrol (CSNP_RES)) was adjusted to 4.0 by the addition of concentrated acetic acid.

Electrospinning solution

The first step was preparation of the electrospinning solution, since the water solution of chitosan cannot be used alone as an electrospun solution, due to its non-stability, low solubility and unsuitable mechanical properties. Therefore, polyethylene oxide (PEO) 5% (*w*/*v*) was added to chitosan nanoparticles’ dispersions (CSNP). The dispersions were prepared with different volume ratios of chitosan nanoparticles, without and with both (poly)-phenols components and polyethylene oxide (PEO) solution. Prior to the electrospinning process, the prepared liquid solutions were characterised, in order to define the optimal volume ratio of liquid formulations and polyethylene oxide (PEO). The conductivity (σ), viscosity (η) and surface tension (γ) of liquid formulations were determined using an electrical conductivity meter (Mettler Toledo, Columbus, OH, United States), viscometer (FUNGILAB, Barcelona, Spain) and tensiometer (Krüss GmbH, Hamburg, Germany). The most optimal CSNP and PEO volumetric ratio was determined, and used for preparation of chitosan nanoparticles’ dispersion solution with encapsulated catechin (CSNP_CAT:PEO) and chitosan nanoparticles with encapsulated resveratrol (CSNP_RES:PEO). The optimisation of the electrospinning procedure followed, varying the processing parameters, such as voltage and distance between electrodes.

#### 2.1.2. Nanofibre Formation

In addition to optimisation of the polymer solution properties, variation of the process parameters and environmental conditions were also performed, as pointed out in Section 3. The electrospinning process was carried out using the NanoSpider NS LAB 500 (Elmarco, Liberec, Czech Republic), applicable for needle-free electrospinning processes. The bathtub contained the spinning electrode, while the cylindrical electrode was used for forming nanofibres. The reposting was also the collecting electrode for newly formed fibres at the same time. In this paper, a chitosan–(poly)-phenols–PEO based solution was electrospun on two different materials: (i) polypropylene material, as standard basic material (Pegatex^®^ S non-woven, kindly supplied by PEGAS NONWOVENS s.r.o., Znojmo, Czech Republic in the form of 100% polypropylene (PP) fibres mesh), and (ii) viscose substrate (VIS), as kindly supplied by producer Tosama d.o.o, Domžale, Slovenia. The polypropylene foundation was used since it is inert, and therefore the elementary composition and morphology was analysed and clarified of the electrospun polymer–(poly)-phenols solution itself. After physico-chemical characterisation, the same formulation was also electrospun onto the viscose substrate, as a representative material used for wound healing. The layer of electrospun chitosan–(poly)-phenols formed on the viscose substrate presented the composite fibrous material, which was then subjected to bioactive testing such as antimicrobial and antioxidant properties, as well as for in vitro release monitoring, to estimate its real potential application. These parameters are crucial for wound healing applications and usage. The electrospun fibrous sample notation and description are listed in Table 1.

### 2.2. Methods for Characterisation of Liquid Formulations

Dynamic light scattering (DLS) and zeta potential determination

The particles’ mean diameter (i.e., hydrodynamic diameter) and the polydispersity index (PDI) of prepared nanoparticles’ dispersion were determined by dynamic light scattering (DLS) (Zetasizer Nano ZS, Malvern Instruments Ltd, Malvern, UK) at a temperature of 25 °C. The zeta potential (ZP) was determined by performing an electrophoresis experiment on the samples, and the velocity of the particles was measured using laser Doppler velocimetry (LDV) on a Zetasizer Nano ZS (Malvern Instruments Ltd, Malvern, UK). Before being analysed, the dispersion was stirred for 15 min and adjusted to pH 4 with acetic acid, if necessary. The results were calculated and presented as the average of three measurements. The standard deviation was calculated by a standard deviation calculator.

Encapsulation efficiency (EE)

The encapsulation efficiency was determined for chitosan nanoparticles containing catechin or resveratrol. The encapsulation efficiency, or entrapment efficiency, was determined as the relation between the concentration of incorporated substance in nanoparticles and the initial concentration of substance in nanoparticles’ dispersion. The concentration of incorporated substances was determined indirectly by spectrophotometric concentration monitoring of a non-incorporated substance in the supernatant after the dispersion of nanoparticles was centrifuged. The monitoring was carried out by the maximum wavelength of each incorporated substance (i.e., catechin and resveratrol). The encapsulation efficiency was evaluated using Equation (1):(1)$$EE=\frac{C_{NP}}{C}=\frac{C-C_{SUP}}{C}\times 100\;\%,$$
where *EE* is the encapsulation efficiency (%), *C* is the initial concentration of substances used for preparation of nanoparticles’ dispersion (g/L), *C_NP_* is the concentration of incorporated substances in the nanoparticles (g/L), and *C_SUP_* is the concentration of substances in the supernatant (g/L).

The results are given as an average of three measurements. The standard deviation was simply calculated using a standard deviation calculator.

Antioxidant efficiency

The antioxidant effectiveness was assessed on the following liquid formulations: chitosan solution (CS), dispersion of chitosan nanoparticles (CSNP), catechin solution (CAT), dispersion of chitosan nanoparticles with incorporated catechin (CSNP_CAT), resveratrol solution (RES) and dispersion of chitosan nanoparticles with incorporated resveratrol (CSNP_RES). A total of 0.1 mL of each sample was added to 3.9 mL of 2,20-azino-bis(3-ethylbenzothiazoline-6-sulfonic acid) (ABTS^•+^) free radicals. The radicals’ inhibition was estimated spectroscopically by measuring the absorbance at 734 nm at 25 °C, in intervals of 15 min and 60 min. The antioxidant effect was evaluated by ABTS^•+^ radicals’ inhibition. The results are presented in percentages and calculated using Equation (2):Inhibition = (A_Control_ − A_Sample_)/A_Control_ × 100,(2)
where A_Control_ is the absorbance measured at the starting concentration of ABTS^•+^, and A_Sample_ is the absorbance of the remaining concentration of ABTS^•+^ in the presence of the applicated chitosan and resveratrol in the solution. The results are given as an average of five measurements. The standard deviation was simply calculated by a standard deviation calculator.

Antimicrobial efficiency

The antimicrobial efficiency of liquid formulations was evaluated using the minimal inhibitory concentration (MIC). The minimal inhibitory concentration of the catechin and resveratrol solution was determined at the Biotechnical Faculty at the University of Ljubljana. The testing was performed using two pathogen microorganisms: *Staphylococcus aureus* (DSM 799) as Gram-positive bacteria and *Escherichia coli* (DSM 1576) as Gram-negative bacteria. The minimal inhibitory concentration was determined using the broth micro dilution method. In total, 50 μL of each bacterial suspension in a suitable growth medium were added to the wells of a sterile 96-well microtitre plate already containing 50 μL of two-fold serially diluted chosen substrate in a proper growth medium. The final volume was 100 μL. Control wells were prepared with a culture medium, bacterial suspension only, chitosan solution, chitosan nanoparticles’ dispersion, resveratrol solution, catechin solution and ethanol in amounts corresponding to the highest quantity present in the resveratrol/catechin solution. The content of each, respectively, was well mixed on a microplate shaker, at 900 rpm for 1 min, prior to incubation for 24 h in the cultivation conditions described by Klancnik et al. [40]. The MIC was the lowest concentration where no viability was observed after 24 h because of metabolic activity [41]. To indicate respiratory activity, the presence of colour was determined after adding 10 μL/well of INT (2-p-iodophenyl-3-p nitrophenyl-5-phenyl tetrazolium chloride, Sigma-Aldrich, London, UK) or TTC (2,3,5-triphenyl tetrazolium chloride) dissolved in water (INT 2 mg/mL, TTC 20 mg/mL), and incubated under appropriate cultivation conditions for 30 min in the dark [42]. To determine the adenosine triphosphate (ATP) activity, the bioluminescence signal was measured by a microplate reader, after adding 100 μL/well of BacTiter-Glo™ reagent and after 5 min incubation in the dark. Positive controls were wells with a bacterial suspension in an appropriate growth medium, and a bacterial suspension in an appropriate growth medium with ethanol, in amounts corresponding to the highest quantity present in the broth microdilution assay. Negative controls were wells with growth medium and chitosan substrates and resveratrol or catechin. The results are presented as average values of three measurements with corresponding standard errors (standard deviation calculation).

### 2.3. Methods for Characterisation of Electrospun Fibrous Samples

The morphology and elemental composition of electrospun chitosan–(poly)-phenols-based solution onto polypropylene material (PP) as standard basic material were analysed. In order to estimate the potential applications of such technology in wound dressings, the same procedure was also applied on the viscose substrate (VIS), as a representative material used in wound healing. The latter composite material was also checked regarding its morphology and, most importantly, its bioactive profile; antimicrobial and antioxidant efficiency were correlated strongly to (poly)-phenols’ release, which was also examined.

Scanning electron microscopy (SEM)

The morphology of electrospun samples (onto PP and VIS) was determined using the scanning electron microscope FE-SEM Supra VP 35, from the company C. Zeiss, Oberkochen, Germany. The machine was equipped with a high-resolution detector for recognition of secondary electrons. It enables the use of variable pressure that, by lower voltage, allows using the high resolution by measuring the samples without any special pre-treatment. Morphology analysis was done for chitosan–(poly)-phenols liquid formulations electrospun onto polypropylene (PP) mesh and viscose (VIS) non-woven, and as such, was analysed using SEM.

Attenuated total reflectance Fourier transform infrared spectroscopy (ATR-FTIR)

The chitosan–(poly)-phenols liquid formulations electrospun onto PP mesh were characterised using Fourier transform infrared spectroscopy (Perkin Elmer FT-IR, Omega, Ljubljana, Slovenia). Analysing the obtained spectra, characteristic peaks were determined, indicating the presence of defined functional groups. The samples were put into a compression diamond cell. The spectra for each sample were recorded 16 times, the measured area was between 4000 cm^−1^ and 650 cm^−1^ and 10 cm^−1^ was used as the interval.

X-ray photoelectron spectroscopy (XPS)

The elemental surface composition of electrospun fibres (onto PP) was carried out by X-ray photoelectron spectroscopy (TFA XPS Physical Electronics, Chanhassen, MN, USA). Spectra were recorded using the XPS instrument TFA XPS (Physical Electronics, Chanhassen, MN, USA) in order to assess the surface of the sample. The base pressure in the XPS analysis chamber was approximately 6 × 10^−8^ Pa. The samples were excited with X-rays over a 400-μm spot area with monochromatic Al Kα1,2 radiation (1486.6 eV) operating at 200 W. Photoelectrons were detected with a hemispherical analyser, positioned at an angle of 45° with respect to the normal sample surface. The energy resolution was about 0.6 eV. Spectra were recorded from at least two locations on each sample, using an analysis area of 400 μm. Surface elemental concentrations were calculated from the survey scan spectra using the Multipak software (PHI MULTIPAK™ VERSION 9.0, Chigasaki, Kanagawa, Japan) as the average of two repeated measurements on each sample. The standard deviation was defined using a standard deviation calculator.

Evaluation of antioxidant properties

The antioxidant effect of the formed electrospun samples (onto viscose) was evaluated spectrophotometrically, using the biochemical ABTS reagent. First, 0.1 g of sample was placed in a vessel and 3.9 mL of diluted ABTS solution poured over it. The inhibition of radicals was determined by measuring the absorbance at 734 nm at 25 °C in intervals of 15 min and 60 min. The antioxidant effect is expressed in percentage, as inhibition of free radicals of the ABTS reagent, as calculated using Equation (2). The results are given as an average of five measurements. The standard deviation was calculated by a standard deviation calculator.

Evaluation of antimicrobial properties

The antimicrobial testing of the produced electrospun samples (onto viscose) was performed at the National Laboratory of Health, Environment and Food in Maribor, Slovenia. Dynamic contact conditions were used according to standard ASTM E 2149-01 [43]. The antimicrobial properties of the prepared samples were tested using *Staphylococcus aureus* (Gram-positive bacteria; DSM 799) and *Escherichia coli* (Gram-negative bacteria; DSM 1576). The results are presented as average values of three measurements with corresponding standard errors (standard deviation calculation).

In vitro release study

In vitro release testing of CAT and RES substances from the electrospun samples (onto viscose) was performed using the Automated Transdermal Diffusion Cells Sampling System (Logan System 912-6, Logan Instruments Corp., New Jersey, NJ, USA). A substance-loaded electrospun sample (square of 1.0 × 1.0 cm^2^) was placed into Franz diffusion cells. The (poly)-phenol substance, catechin or resveratrol, was released into the phosphate-buffered saline (PBS) medium with a pH of 7.4. Samples were collected at different time intervals over a period of 24 h (5 min, 10 min, 20 min, 30 min, 60 min, 120 min, 180 min, 240 min, 300 min, 360 min and 1440 min). The concentration of released catechin or resveratrol from electrospun samples was determined using a UV–Vis spectrophotometer (Agilent Cary 60 UV-Vis Spectrophotometer, Agilent. Com., Braunschweig, Germany) by quantification of the absorption peak at 279 nm for catechin and 313 nm for resveratrol.

The withdrawn volume of samples was replaced by fresh ultrapure water. This dilution was accounted for in the calculation of the concentration using the Beer–Lambert law (based on a previously prepared calibration curve for the active substance). The in vitro release study was performed in three parallels for each prepared electrospun sample, and the results are presented as average values with standard errors.

All release studies were performed in (at least) three parallels. For all obtained release results (presented as part of Figures 5–7 and as a part of the Appendix A), the confidence interval was determined as ±ts/√x, where t is a Student’s t-distribution, s is the standard deviation and x is the number of measurements

## 3. Results and Discussion

### 3.1. Characterisation of Liquid Formulations

#### 3.1.1. Particle Size, PDI and Zeta Potential Determination

The size of the synthesised nanoparticles’ hydrodynamic diameter ($d\bar{H)}$, zeta potential (ZP) and polydispersity index (PDI), were determined using the DLS method. The analysed results of dispersion of chitosan nanoparticles (CSNP), dispersion of chitosan nanoparticles with incorporated catechin (CSNP_CAT) and dispersion of chitosan nanoparticles with incorporated resveratrol (CSNP_RES), are shown in Table 2.

The average size of chitosan nanoparticles was 379.7 nm. In both cases of modified nanoparticles (i.e., CSNP_CAT and CSNP_RES), an increase was observed in the hydrodynamic diameter. For CSNP_CAT, the average size was 2986.7 nm, and for CSNP_RES, the average size was 1874.5 nm. Increasing the size of chitosan nanoparticles with an incorporated active substance ((poly)-phenols) from the nano to micro scale confirmed the binding of the active ingredients to the interior, or onto the surface of the chitosan nanoparticles. The high PDI value in all cases of particles suggests that there was a wide size distribution of particles. The latter may also indicate the presence of agglomerates in the dispersion. For CSNP_CAT, this claim confirmed the value of the zeta potential, which indicates the instability of the dispersion and the tendency of particles to agglomerate. For CSNP and CSNP_RES, the value of the zeta potential was higher than 30 mV. Above this value, in accordance with the theory [44,45], electrostatic repulsive forces predominate among particles, thereby preventing agglomeration of particles. In all cases, the pH of the dispersion was approximately 4. At this value, the amino groups of both chitosan particles and chitosan particles with an incorporated active substance are protonated (−NH_3_^+^), which provides the same positive charge in the dispersion, thus allowing electrostatic repulsive forces sufficiently to lead to the value of the zeta potential of more than 30 mV. In the case of catechin binding (CSNP_CAT) the amine groups were less accessible or blocked; thus, due to the lower number of protonated amines, the repulsion forces were reduced, and the resulting agglomeration of particles occurred.

#### 3.1.2. Encapsulation Efficiency

The encapsulation efficiency (EE, %) of catechin and resveratrol in chitosan particles was determined using UV–Vis spectroscopy, as explained in the experimental part. Using Equation (1), the EE values were calculated, and are presented in Table 3.

The encapsulation efficiency depends largely on the active ingredient, the chosen analytical method and the conditions for the preparation of particles. The obtained results show better efficacy of resveratrol encapsulation in chitosan particles (63%) compared to catechin entrapment in chitosan particles (11%). The difference in the encapsulation efficiency may be due to the structure of the selected (poly)-phenolic active substance, or different input concentrations of the (poly)-phenolic active substance. The particles were prepared from a solution of resveratrol with a weight concentration of 20 g/L, while the outlet mass concentration of the catechin solution was 10 g/L. Obviously, a higher concentration of (poly)-phenols makes it possible to increase the availability and trapping of (poly)-phenols into particles. Comparing catechin and resveratrol from the structure point of view, catechin, due to tricyclicity, higher molecular weight (290.27 g/mol) and branching structure, and thus possible steric obstruction, does not show such a high affinity for capture in chitosan particles.

#### 3.1.3. Antioxidant Efficiency

The effect of antioxidant properties of the prepared liquid formulations (chitosan solution, CSNP, catechin solution, CSNP_CAT, resveratrol solution, CSNP_RES) was evaluated using a spectrophotometric method, based on the use of the biochemical reagent ABTS. This method monitors the change in the concentration of the radical ABTS^•+^, which occurs in the presence of an antioxidant. The antioxidant effect (as % antioxidant inhibition) of the prepared chitosan solution and the dispersions of chitosan particles alone, as well as with entrapped catechin or resveratrol after 15 and 60 min, is shown in Figure 1.

Results in Figure 1 show that the reduction of the ABTS^•+^ radicals of the chitosan solution (CS) and the dispersion of chitosan particles (CSNP) after exposure was very low at both time intervals. After 15 min, it can be seen that neither CS nor CSNP liquid formulations showed the inhibition of free radicals, while, after 60 min of exposure, the inhibition increased slightly (i.e., 9.8% for CS and 5.5% for CSNP). Based on the obtained results, both the chitosan solution and the dispersion of chitosan particles did not show any strong antioxidant effect, even after a 60 min time interval.

The catechin solution showed antioxidant effect already after 15 min of monitoring, while the inhibition of the ABTS^•+^ increased further within 60 min. At the beginning, the catechin entrapped in chitosan particles showed less antioxidant activity, while, after 60 min, it increased up to 75% due to its release. By using 10-times lower concentration (i.e., 0.05 g/L) of both liquid formulations (not shown here), surprisingly catechin encapsulated into chitosan particles showed better antioxidant properties (i.e., 50% within 15 min and 80% within 60 min) compared to catechin solution (i.e., 39% within 15 min and 66% within 60 min). This suggests a synergistic effect between chitosan and catechin. The catechin antioxidant properties were preserved after incorporation in particles, while the chitosan worked additively with small antioxidant capacity, and, in addition, increased the antioxidant potential of catechin itself.

Using different concentrations (0.05 g/L, 0.1 g/L, 0.5 g/L) when preparing resveratrol and resveratrol particles’ liquid formulations, the antioxidant inhibition results (not shown here) indicated dependency between the inhibition of radical ABTS^•+^ and the liquid’s concentration (i.e., increase of inhibition by increasing the liquid’s concentration). In addition, taking into account all liquid formulations, the highest antioxidant effect was observed by the resveratrol solution within 15 min and within 60 min of monitoring. The effect was also preserved after the resveratrol inclusion into chitosan particles, also pointing out the synergistic effect between chitosan and resveratrol, but in this case, more pronounced.

The experiment suggests that catechin and resveratrol are either released from the chitosan particles, and/or that they are readily available on particles’ surfaces. Independent of time, it may be seen that chitosan nanoparticles with embedded resveratrol showed better antioxidant activity than particles with embedded catechin. The latter may be connected to a lower initial concentration of catechin in comparison to resveratrol. A one-time higher concentration of resveratrol was chosen for the encapsulation. Different initial concentrations are the consequence of different solubility of (poly)-phenols into water. The upper limit was chosen for both cases.

#### 3.1.4. Minimal Inhibitory Concentration (MIC)

Table 4 summarises the minimal inhibitory concentration (MIC) for the chitosan (CS), chitosan nanoparticles (CSNP), catechin (CAT) and resveratrol (RES) solution, which was determined by the dilution method in the liquid medium. The test was carried out for the pathogen microorganisms *Staphylococcus aureus* and *Escherichia coli*. Changes of colour were observed and recorded. The lowest concentration prior to colour change was considered as the minimum inhibitory concentration (MIC). The smaller the MIC value, the greater the antimicrobial activity of the selected active substance.

The MIC of the resveratrol solution for *Escherichia coli* was 5 mg/mL and 0.156 mg/mL for *Staphylococcus aureus*. The solution of catechin, compared to the resveratrol solution, exhibited slightly less antimicrobial activity, since the MIC values were 10 mg/mL for *Escherichia coli* and 2.5 mg/mL for *Staphylococcus aureus*. For both liquid formulations, the MIC results showed that the (poly)-phenolic active substances resulted in more effective antimicrobial activity against *Staphylococcus aureus*, which is a Gram-positive bacterium. The latter could be supported by in vitro studies, showing that Gram-positive bacteria are considered to be more sensitive to plant phenolic agents, compared to Gram-negative bacteria [46,47]. Similar findings, regarding the efficiency against Gram-positive bacteria, were also pointed out for chitosan [48]. The exact data of MIC for chitosan macromolecular solution as well as chitosan nanoparticles were published and discussed by Glaser et al. [49].

#### 3.1.5. Electrospinning Process Parameters

The dispersions with different volume ratios of chitosan particles were prepared without and with both (poly)-phenolic components and polyethylene oxide (PEO) solution. Used prior in the electrospinning process, the prepared liquid solutions were characterised in order to define the optimal volume ratio of liquid formulations and polyethylene oxide (PEO). The determined physical properties of liquid formulations and PEO are listed in Table 5. The results are given as average of three measurements along with the standard deviations of the mean.

The results show that in order to achieve smooth electrospinning, the most optimal ratio for chitosan particles without and with imbedded (poly)-phenols and polyethylene oxide, was identified as CSNP/CSNP_CAT/CSNP_RES:PEO = 2:1, while for solutions containing only (poly)-phenols, the ratio was CAT/RES:PEO = 1:2. These volume ratios were then used further for preparing the electrospinning solutions, used for producing nanofibres.

Variations of the process parameters and environment conditions were also performed, alongside optimisation of the polymer solution properties. The intensity of the electric field was performed via regulation of the applied voltage, rotation of the roller electrode (containing the spinning solution) and the distance between the roller electrode and collector electrode (PP mesh or viscose non-woven for collecting nanofibres). Environmental parameters were also adjusted, like humidity (RH) and temperature (T), since they also influence the forming of nanofibres. According to the experimental trials, supported by the morphology results of the formed nanofibres obtained by scanning electron microscopy (not presented here), the optimal electrospinning parameters were defined as follows: voltage = 60 kV, rotation speed of roller electrode with solution = 3.8 m/min, distance between roller and collector electrode = 150 mm, RH = 35 ± 5%, T = 20 ± 2 °C.

### 3.2. Characterisation of the Prepared Nanofibres

#### 3.2.1. Morphological Properties of Functional Nanofibres

The morphological properties of the electrospun fibres were characterised using scanning electron microscopy (SEM). The SEM results of liquid formulations of CSNP:PEO = 2:1, CAT:PEO = 1:2, RES:PEO = 1:2, CSNP_CAT:PEO = 2:1 and CSNP_RES:PEO = 2:1, electrospun on a collecting substrate, are presented in Figure 2, using different magnifications. The effect of the formed nanofibres was analysed on the supporting substrate material (i.e., on polypropylene mesh (left column), and on viscose non-woven (right column), at 500× magnification).

One can see that the SEM figure of polypropylene substrate (PP) shows a smooth and very regular surface. The figure also shows an interweave of polypropylene fibres forming the mesh structure. The surface morphology of the reference viscose substrate (VIS) is smooth with no significant surface structural features, except the parallel grooves that run alongside the fibre axis, which originate from the production process.

The SEM figures, using the liquid formulation of CSNP:PEO, evidence the individual thicker spheres and irregular layers of the deposited polymer solution. The latter may indicate the presence of chitosan particles or their agglomerates, as also evidenced by PDI (see Table 2). Taking into account the measured physical properties of CSNP and CSNP:PEO solutions (see Table 5), the addition of PEO lowered the conductivity (by approx. 30%) and increased the viscosity significantly (more than 10 times), while the surface tension remained almost the same. The physical parameters of the CSNP:PEO solution were the reason for the incomplete formation of a Taylor cone; thus, reduced extension of the polymer solution via forming fibres could occur during the electrospinning. Taking into account the collecting material for depositing the nanofibres, the SEM images indicate different affinity of nanofibres to cover the substrate material. For the polypropylene substrate material, the formed nanofibres also overlapped the space between the substrate material fibres. Having the viscose substrate material, the formed nanofibres were deposited only onto fibres already present in the substrate.

Using the electrospinning solution containing only (poly)-phenols (i.e., CAT:PEO and RES:PEO), the SEM figures evidenced no formation of nanofibres, regardless of the used substrate. The empty spaces within the PP mesh and viscose fibres of the non-woven were filled with polymer solution. In addition, some segmentation could be observed, indicating some separate areas of segments pointing out the fibrous structure. The non-success in forming fibres resulted from the physical properties of the used solution and molecular weight of the substrates. The results presented in Table 6 evidence that both solutions contacting only (poly)-phenols have lower conductivity (96%) and surface tension (42%), while much higher viscosity (10 times higher) compared to the chitosan particles’ solution. Moreover, the reason may also be the low molecular weight of catechin (290.27 g/mol) as well as resveratrol (288.25 g/mol).

The SEM images of the electrospun CSNP_CAT:PEO and CSNP_RES:PEO samples showed only the formation of a thin polymer nanofilm on both used substrates (PP, VIS). The influence of the polymer solution affinity to the fibre-substrate was observed again. The space between the polypropylene mesh was not filled, while the fibres of the viscose substrate material were covered with the polymer–(poly)-phenols thin film. There was no evidence of nanofibre formation. Comparing the physical properties of the prepared solutions, all measured parameters for chitosan particles with both (poly)-phenols were lower compared to the solution without catechin and resveratrol. The conductivity was reduced by 60%, the surface tension by 33%, while the lowest reduction was obtained by viscosity (3% for catechin and 16% for resveratrol).

In summary, one could conclude that (poly)-phenols obviously have significant impact on the physical properties of solutions that affect the nanofibre forming process. The main influence the added (poly)-phenols had was on conductivity, which they reduced from 60% up to 96%. Thus, the lack in forming nanofibres (regardless of the used substrate) was attributed to relatively low electric conductivity, since higher conductivity would enable carrying more electrical charges during the electrospinning process. In addition, by having solutions with high-charge density, effective stretching of the Taylor cone through repulsion of one sign charges would occur [50].

The SEM figures of chitosan particles with and without (poly)-phenols, indicate rather on the electro (spraying) coating process and formation of their films onto the existing fibre structure of the basic material. However, when whole composite fibrous materials are analysed, they may be evaluated as nanofibres, whilst at least one of the dimensions of the composite fibrous structure is on the nano scale.

#### 3.2.2. Chemical Composition of Functional Nanofibres

The chemical structure of the liquid formulations electrospun onto a polypropylene (PP) substrate was determined using attenuated total reflectance Fourier transform infrared spectroscopy (ATR-FTIR). In Figure 3, the spectra of basic substances as references—chitosan (powder), catechin (powder), resveratrol (powder), and polyethylene oxide (PEO, powder)—were recorded, and compared to the spectra of CSNP:PEO = 2:1, CSNP_CAT:PEO = 2:1 and CSNP_RES:PEO = 2:1 samples, prepared by the process of electrospinning on a polypropylene substrate. In addition, the spectra of polypropylene substrate were also recorded, as the basic vehicle material.

The characterisation of the PEO sample (as powder) was carried out using ATR-FTIR apparatus. The spectra show that transmittance peaks were observed at 1253 cm^−1^ which are characteristic for an ethereal group (C–O–C). In addition, a stretching band between 3300 cm^−1^ and 3400 cm^−1^ appeared pointing out the -OH groups [51].

The ATR-FTIR spectrum of chitosan shows peaks overlapping in the range between 3600 cm^−1^ and 3200 cm^−1^ characteristic for the hydroxyl group (–OH) and amine group (–NH_2_). The peak appearing at the wavenumbers 2921 cm^−1^ and 2869 cm^−1^ corresponds to the C–H tensile vibrations (symmetric and asymmetric stretching, respectively). The peak at 1653 cm^−1^ belongs to the C=O stretching and the peak at 1584 cm^−1^ to the N-H stretching and ether bond. At the wavenumber 1420 cm^−1^ is a peak characteristic for the CH–OH bond and at wavenumber 1375 cm^−1^ is a peak for CH_2_–OH. The transmittance peak at 1150 cm^−1^ corresponds to a non-symmetric stretch of the C–O–C bridge. The transmittance peaks at 1078 cm^−1^ and 1026 cm^−1^, respectively, are present due to the skeletal vibration characteristic for the C–O bond [49,52].

The catechin spectrum shows a strong peak at wavenumber 3221 cm^−1^ characteristic for the tensile vibration of the hydroxyl group. The transmittance peak at wavenumbers 1607 and 1518 cm^−1^ corresponds to vibrations of the C=C alkene bond and C=C aromatic bond, respectively. The transmittance peak at 1457 cm^−1^ belongs to the C–H bond characteristic for alkanes. The signals at the wavenumbers (1282, 1141, 1019 and 962 cm^−1^) can be attributed to C–O alcohols, –OH aromatic, –C–O alcohols and C–H alkanes, respectively, as found in the catechin structure [53].

The ATR-FTIR spectrum for resveratrol shows a transmittance peak at the wavenumber 3180 cm^−1^ which is typical for the tensile vibrations for the O–H bond. The peak observed at 1604 cm^−1^ fits to the expansion of the aromatic C=C double bond. At the wavenumber 1583 cm^−1^ is a peak for the alkene (specific bonding of the C–C bond occurred), while at 1380 cm^−1^, is peak typical for bond vibration of phenolic O–H. Transmittance peak at wavenumber 1146 cm^−1^ belongs to the stretching of phenolic C–O bond. Peaks at the wavenumbers 964 cm^−1^ and 827 cm^−1^ correspond to the =C–H band. Transmittance peak at 964 cm^−1^ corresponds to =C–H band of alkenes in trans-resveratrol, while transmittance peak at 827 cm^−1^ belongs to =C–H band of arenes conjugated to the olefinic band [49,54].

The spectrum of the sample PP_CSNP:PEO = 2:1 shows transmittance peaks characteristic for chitosan. At the wavenumber 3352 cm^−1^ is a strong peak, typical for the hydroxyl (–OH) and amine groups (–NH_2_). The transmittance peak at wavenumbers 2954 cm^−1^ and 2876 cm^−1^ corresponds to the symmetric and asymmetric stretching of the C–H bond. In addition, transmittance peak located at wavelength 1638 cm^−1^ agrees to C=O stretching and a peak at 1559 cm^−1^ corresponds to N-H stretching and ether bond.

The ATR-FTIR spectrum of the electrospun sample PP_CSNP_CAT:PEO = 2:1 shows characteristic peaks for chitosan. Transmittance peaks at wavenumbers 2919 and 2876 cm^−1^ correspond to the C–H bond (symmetric and asymmetric stretching). The transmittance peaks at wavenumbers 1375 cm ^−1^ and 1342 cm ^−1^ point out the bonds CH–OH and CH_2_–OH, respectively. The signals at the wavenumbers 1145 cm^−1^ and 1102 cm^−1^ correspond to the skeletal vibration of the C–O bond. Peaks at wavenumbers 1281 cm^−1^ and 961 cm^−1^ can specify to the presence of catechin. In addition, peak at wavenumber 1281 cm^−1^ can be attributed to C–O bond characteristic for alcohols and peak at wavenumber 961 cm^−1^ to C–H bond characteristic for alkanes (catechin).

The ATR-FTIR spectrum of the electrospun sample PP_CSNP_RES:PEO = 2:1 displays characteristic transmittance peaks for the chitosan. The transmittance peaks at the wavenumbers 2919 cm^−1^ and 2867 cm^−1^ correspond to the C–H tensile vibration (symmetric and asymmetric stretching). Transmittance peak at 1588 cm^−1^ indicates the C=O stretching. Transmittance peak at 1513 cm^−1^ belongs to the N–H stretching and the ether bond. The peaks at the wavenumbers 1148 cm^−1^ and 1103 cm^−1^ agree to the skeletal vibration of the C–O bond. Transmittance peak at wavenumber 1375 cm^−1^ can be attributed to phenolic O-H bond from resveratrol. In addition, smaller peaks at the wavenumbers 972 cm^−1^ and 827 cm^−1^ could indicate the presence of the olefinic band, which is also characteristic for resveratrol.

#### 3.2.3. Surface Elemental Composition of Functional Nanofibres

The elemental surface composition of the electrospun formed nanofibres on the polypropylene substrate was determined using X-ray photoelectron spectroscopy (XPS). The results are presented in Table 6.

The polypropylene substrate was used as a reference sample, upon which the surface composition of the prepared electrospun samples could be discussed. The increased percentage of carbon was found in almost all electrospun samples, except for PP_CSNP:PEO = 2:1, where the percentage was lower (by 6%) compared to the reference polypropylene mesh. Comparing samples prepared using chitosan particles’ solution (PP_CSNP:PEO = 2:1) and only the (poly)-phenols solutions (PP_CAT:PEO = 1:2; PP_RES:PEO = 1:2), the increase in atomic % of C was more expressed by catechin (by 20%) compared to resveratrol (6.6%), while the increase was even more significant in samples containing (poly)-phenols embedded in particles (PP_CSNP_CAT:PEO = 2:1 and PP_CSNP_RES:PEO = 2:1), indicating the same trend (i.e., an increase of 26.6% for catechin and 10.6% for resveratrol).

The increased amount of atomic % of oxygen was shown by samples containing chitosan particles and chitosan particles with embedded resveratrol compared to the reference polypropylene substrate. Comparing chitosan particles with (PP_CSNP_CAT:PEO = 2:1 and PP_CSNP_RES:PEO = 2:1) and without (poly)-phenols (PP_CSNP:PEO = 2:1), one can see that (poly)-phenols reduced the atomic % of oxygen. The effect was more pronounced by catechin (i.e., decreased by 42.7%) compared to resveratrol (decreased by 19.4%). The same trend was observed also for (poly)-phenols solutions alone (i.e., a higher decrease by catechin (by 37.1%) and lower by resveratrol (by 21.8%)) compared to PP_CSNP:PEO = 2:1 sample.

The increase in atomic concentration of nitrogen was evident in the PP_CSNP:PEO = 2:1, PP_CSNP_CAT:PEO = 2:1 and PP_CSNP_RES:PEO = 2:1 samples, compared to the polypropylene substrate. The latter indicated that the formed nanofilm contained chitosan, trapped in particles, was detected on the surface. The comparison between the samples with/without catechin and resveratrol, indicated a decrease in the amount of nitrogen present on the surface of PP_CSNP_CAT:PEO = 2:1 and PP_CSNP_RES:PEO = 2:1 samples, compared to PP_CSNP:PEO = 2:1. The same trend (i.e., higher decrease of the atomic % of nitrogen) was observed for catechin (by 52%), and lower for resveratrol (by 38%) compared to chitosan particles without (poly)-phenols. The results pointed out that (poly)-phenols, during their incorporation into chitosan particles, reacted with chitosan and even, most probably, covered the chitosan particles. Regarding the latter, the chitosan lost its accessibility on the thin layer of the surface and, consequently, the amount of surface nitrogen detected is reduced.

In general, it may be concluded that chitosan particles, due to amino groups with embedded antioxidants, are available on the matrices’ surfaces, which is important for release into contact media (into wounds by real applications).

#### 3.2.4. Antioxidant Properties of Functionalised Nanofibres

Based on the spectrophotometric monitoring of the change in the concentration of free radicals of the ABTS biochemical reagent, the inhibition of free radicals was calculated according to Equation (2). Results, showing the antioxidant effect of CSNP:PEO = 2:1, CSNP_CAT:PEO = 2:1 and CSNP_RES:PEO = 2:1 liquid formulations, electrospun onto viscose samples, are shown in Figure 4.

From the graphic representation of inhibition results (Figure 4), it can be seen that the sample with chitosan particles electrospun onto viscose (VIS_CSNP:PEO = 2:1) does not exhibit antioxidant properties. Inhibition of radicals ABTS^•+^ amounted to 21.8% after 15 min and 38.3% after 60 min. As expected, the samples with both (poly)-phenols, electrospun onto viscose substrate (VIS_CSNP_CAT:PEO = 2:1 and VIS_CSNP_RES:PEO = 2:1), showed very good inhibition of ABTS^•+^ radicals. The antioxidant effect for the sample VIS_CSNP_CAT:PEO = 2:1 had changed very little over the time interval, such that after 15 min and 60 min it showed 95.2% and 95.5% inhibition, respectively, of the ABTS^•+^ radicals. However, the sample VIS_CSNP_RES:PEO = 2:1 achieved better antioxidant effect after 15 min (96.6%) as compared to sample VIS_CSNP_CAT:PEO = 2:1, and above all, after 60 min (88.4%). Summarising, both fibrous samples with incorporated (poly)-phenols clearly expressed high antioxidant effectiveness, whereas the latter persisted longer with catechin, which has to be taken into account in the explanation of the antimicrobial properties, as well as the in vitro drug release performance of the prepared samples. It is known that the wound healing process can be supported by the presence of antioxidants, therefore it is of great importance to incorporate these properties into the wound healing concept. The general role of antioxidants appears to be important for the successful treatment and management of wounds. Antioxidants reduce these adverse effects of wounds by removing inflammatory products. They counteract the excess proteases and reactive oxygen species often formed by the accumulation of neutrophils at the injured site and protect protease inhibitors from oxidative damage. The most likely mechanism of antioxidant protection is the direct interaction of the extracts (or compounds) and hydrogen peroxide rather than altering cell membranes and limiting damage. Compounds with high radical scavenging capacity have been shown to facilitate wound healing [55]. Thus, produced nanofibres, VIS_CSNP_RES:PEO and VIS_CSNP_CAT:PEO, are extremely promising regarding this kind of action.

#### 3.2.5. Antimicrobial Properties of Functionalised Nanofibres

The antimicrobial testing of composite viscose nanofibrous material was performed according to ASTM E 2149-01, under dynamic contact conditions. Table 7 shows the reduction (R, %) of electrospun samples after exposure to pathogen microorganisms (i.e., to *Escherichia coli* for 1 h and 6 h, and to *Staphylococcus aureus* for 1 h). The results are given as the average of three measurements.

The materials under investigation confirmed antimicrobial effectiveness as soon as the obtained reduction (R) exceeded 75%. The electrospun samples containing chitosan particles (VIS_CSNP:PEO = 2:1) and chitosan particles with incorporated (poly)-phenol substances (VIS_CSNP_CAT:PEO = 2:1 and VIS_CSNP_RES:PEO = 2:1), showed similar and excellent antimicrobial efficiency toward the *E. coli* microorganism used, regardless of exposure time. Taking into account the testing with *S. aureus*, the chitosan particles and chitosan particles with incorporated (poly)-phenol substances, showed (Table 7) sufficient antimicrobial efficiency, but the reduction was lower compared to the values obtained by exposure to *E. coli*, which is surprising according to the MIC results of pure (poly)-phenols.

With the view of comparing the exposure times (i.e., the samples being in contact with the bacteria for 1 h or 6 h), the results of reduction showed that, in the case of *E. coli*, samples were even more effective after prolongation of time. The latter suggests that some time is needed before the antimicrobial efficiency is being noticed due to the release of bioactive substances, as pointed out below in Section 3.2.6. The results of the reduction show that in the case of *E. coli,* the samples were effective after only 1 h of exposure, resulting in 99.9%, while in the case of *S. aureus* the result was still efficient but less optimistic to continue with an exposure time of up to 6 h. Moreover, this was done only for the bacteria *E. coli*, whilst this bacterium is more complicated and resistant. However, a 6 h test is not necessary, whereas the standard requires a test after 1 h of exposure.

Electrospun samples, by using catechin and resveratrol substance alone, dissolved in PEO solution (VIS_CAT:PEO = 1:2 and VIS_RES:PEO = 1:2), did not show antimicrobial activity, either on *E. coli* or on *S. aureus*, since the reduction was much below the limit. The reduction on *Escherichia coli* in both time frames (i.e., exposure of 1 h and 6 h) was below 40%, while for *Staphylococcus aureus*, even a negative reduction was obtained, indicating stimulation growth of bacteria rather than inhibition. It seemed that too low of (poly)-phenols concentration was embedded. From this it may be concluded that poly/phenols alone do not have any significant influence on antimicrobial activity which may be connected to a low concentration profile for antimicrobial efficiency. Obviously, in this synergistic formulation chitosan behaviour dominated regarding in this synergistic formulation regarding antimicrobial reduction behaviour.

Both bacteria, Gram-positive bacteria *Staphylococcus aureus*, and Gram-negative bacteria *E. coli* were strongly inhibited by prepared nanofibres. The antimicrobial activity, which was only achieved with chitosan, was retained, while the (poly)-phenolic substances enclosed in the chitosan also possessed additional antioxidative properties. Both simultaneous activities can lead to improved wound healing. Among several bacterial wound isolates in the European Union (EU), *Staphylococcus aureus* was the predominant bacterium, followed by *Escherichia coli*. Among the *S. aureus* isolates, 60.6% were Methicillin-resistant Staphylococcus aureus (MRSA)strains. MRSA, a leading strain of wound infection, affects significant areas of the skin or deeper soft tissues such as abscesses, cellulitis, burns or infected deep ulcers. Extended spectrum β-lactamases (ESBLs) producing *Enterobacteriaceae* are also at the forefront of wound infections. In ESBL, positive strains of plasmid-mediated AmpC enzymes and carbapenem, which hydrolyses β-, have given lactamase (carbapenemases) resistance to the newer β- lactam antimicrobials. ESBLs have been most commonly detected in *Escherichia coli* and *Klebsiella* spp., including other bacterial species such as *Salmonella enterica*, *P. aeruginosa* and *Serratia marcescens*. This increase in antimicrobial resistance further delays wound healing and the infection becomes worse, prolonging hospital stays, prolonging trauma care and causing high medical costs [56].

The nanofibres we developed here, can therefore have a positive effect on drug-resistant pathogens and in this way bring wound infections under the control.

#### 3.2.6. In Vitro Release Study

An important aim of this study was to prepare a multi-functional medical textile material, to be preferably used in wound care. Considering the applicability of such materials in the mentioned field, it was very important to evaluate the influence of the prepared formulations by monitoring the release of the two incorporated active agents (CAT and RES) within the simulated wounded skin environment. Based on the usual frequency of changing wound dressings of the most common wounds, which is approximately two days [57,58], it was our purpose to adjust the formulation to enable a sustained release for the mentioned period.

To discuss the results of the performed in vitro release study best, we show these in three diagrams:-The concentration of the released substances as a function of time (Figure 5),-The cumulative mass of the released drug as a function of time (Figure 6),-The percentage of the released drug as a function of time (Figure 7).

These different representation types complement each other in terms of predicting the applicability of the prepared materials (as discussed in the experimental section, to best simulate an actual clinically used wound dressing, the nanofibres were for testing purposes collected on a viscose substrate (VIS)) in an actual clinical setting. The concentration-based representation enabled the evaluation of potential concentration variations, which, on the one hand, corresponded to the specific formulation related properties (e.g., formulation stability, multi-step release due to differently soluble phases in the sample, etc.), and on the other, presented a good indication of the fluctuations in the concentration at the application site, which influence the therapeutic efficiency of the prepared formulation. The other two representations served as the basis for evaluation of the release mechanism, as well as enabled drawing conclusions about the formulations’ ability to tailor to specific patient needs (i.e., in terms of treatment time and time of effect onset, acute wounds require immediate effects, while the onset is not that important in the case of chronic wounds).

Figure 5 shows the changes in release of CAT and RES concentrations as a function of time. Most of the incorporated substances (true for both CAT and RES) from the electrospun samples (VIS_CSNP_CAT:PEO = 2:1 and VIS_CSNP_RES:PEO = 2:1) were released within the first 360 min, which was the case despite the 10-times higher solubility of CAT in water, compared to RES.

Nevertheless, this difference in the solubility was most likely the main reason why CAT reached its peak concentration (0.00762 mg/mL) in the release medium (based on PBS with a pH of 7.4) already after 10 min of the release. The release profile of sample VIS_CSNP_CAT:PEO = 2:1 showed a very steep increase in the first 10 min, after which the concentration started to decrease until reaching a plateau.

The release profile of RES is clearly quite different compared to the one for CAT. However, after a more detailed observation, these differences made sense. Namely, as already hypothesised above, the difference in solubility of both incorporated active ingredients seems to have had an important influence on their release. For the sample VIS_CSNP_RES:PEO = 2:1 the release profile initially showed a “much” slower (compared to CAT) increase in the released concentration. The peak concentration was not reached until 180 min (0.00986 mg/mL), when the concentration of the incorporated substance was decreasing slowly. A full plateau in RES concentration was not reached until 1440 min.

Comparing the obtained release results for CAT and RES further, it is also important to consider potential interactions that can occur between the primary formulation component (chitosan) and the incorporated active ingredients. First, considering the molecular structure of CAT and RES, the reason for a higher solubility of the former becomes clear immediately. CAT has two more OH groups per molecule, and lacks the aliphatic bridge between the phenol rings, both affecting the respective solubility importantly. Now, coming back to the potential interaction with chitosan. Considering the polarity of the latter in the context of the number of OH groups in CAT and RES, as well as in relation to the release medium pH (7.4), one of the possible additional explanations of the differences in the release profiles might be electrostatic repulsion between partially dissociated functional groups. The latter is higher in the case of CAT, with a higher number of OH groups per molecule. Two additional phenomena need to be considered here as well, the first being the low solubility of chitosan at the mentioned pH, as well as its swelling. Both together can affect the release mechanism of both active ingredients, which is most likely governed by a diffusion-controlled process, whereas the swelling contributes partially to the prolongation of RES release, which “sticks” longer to the formulation (due to the above-mentioned lower repulsion). As already mentioned above, RES and CAT molecules were most likely not encapsulated solely into the formulations’ (sphere) interior but were either entrapped in the shell of the particles or “attached” to their surfaces. Finally, an important parameter to be considered in the explanation of the release profiles is also the entrapment efficiency, which is significantly higher in the case of RES. Therefore, it might be the case that, for the latter, a higher portion of the entrapped substance had to cross the chitosan particle shell, which took longer due to the already mentioned reasons.

Figure 6 shows the cumulative masses of the released CAT and RES, as a function of time from the prepared electrospun samples (VIS_CSNP_CAT:PEO = 2:1 and VIS_CSNP_RES:PEO = 2:1). It further includes the calculated first derivatives of the obtained release results for respective substances, which were prepared to shed more light during evaluation of the CAT and RES release mechanisms.

There are several important observations about the release result representation shown in Figure 6. Firstly, we can observe that, despite the differences between CAT and RES, as well as their interaction with chitosan, both substances reached similar release profiles after 360 min. Further, both substances were released from the samples (despite their influence on the electrospinning solution properties and their different entrapment efficiencies) in similar final amounts (after 1440 min). Considering the above release profile, after 1440 min approximately 0.074 mg of catechin (from sample VIS_CSNP_CAT:PEO = 2:1) and approximately 0.072 mg of resveratrol (from sample VIS_CSNP_RES:PEO = 2:1) were released from the prepared electrospun samples. This result is also important for the application potential of the prepared materials. Namely, we now know that initially, with the increasing released concentration of the active substances until 360 min of exposure, a highly efficient antioxidative effect can be achieved, which is then, with the constant release, and hence antioxidative effect, retained for samples with both incorporated active substances until at least 24 h of release. Nevertheless, there are also important differences that are clear from Figure 6. The respective solubility and interaction potential with chitosan also plays an important role here in affecting the initial released masses of CAT and RES. In a similar fashion to the above described results, the higher solubility and potential repulsion with chitosan, led to a faster increase in the release rate for CAT compared to RES.

Considering both main deductions described for the results shown in Figure 6 (similar release after 360 min; differences in the released masses and rates within 360 min) now in an actual clinical application, we can claim that we prepared a formulation that enables us to provide an antioxidant effect either immediately (in the case of incorporated CAT), or with a slower increasing activity until 3 h after application. Combining the antioxidant effects of CAT and RES with the antimicrobial activity of chitosan, makes the prepared formulation more promising for treatment of infected and inflamed wounds; whereas chitosan affects the former positively, CAT and RES with their antioxidant effects, affect the latter.

As mentioned above, to shed some more light on the release mechanism, we also prepared first derivatives of the release data. In general, this enabled us to determine three release regions. The initial region shows a very fast release (until 30 min), the second is considered as a slower release rate, which can nevertheless be still considered as relatively fast (from 30 min up to 360 min), and the final is viewed as a slow release region, where the plateau in the release is reached (from 360 min up to 1440 min). No more substance was released after the first day of release (data not shown; we also sampled the release after 2880 min, but the released CAT and RES masses did not change any more).

The final representation of the release data is shown in Figure 7, which shows the % of the released CAT and RES substances. This representation complements the one shown in Figure 6. Namely, it enables an even faster evaluation of the released CAT and RES amounts without considering their absolute released masses.

From Figure 7, we observe that approximately 90% of CAT and RES were released in the first 360 min. The actual percentage of released substance from the electrospun material was determined based on the initial concentration and encapsulation efficiency of the selected substance. Considering the latter, we calculated that 31% of catechin was released (from sample VIS_CSNP_CAT:PEO = 2:1) and 3% of resveratrol (from sample VIS_CSNP_RES:PEO = 2:1). The remaining amount of CAT and RES is chemically bonded with the chitosan or substrate material (viscose non-woven). As already mentioned above, also here, the main “culprit” for this relatively low release efficiency related to the respective component solubility (and, hence, interaction potentials) in the used release medium (PBS with a pH of 7.4).

Finally, this resulting representation type was used further to fit with some commonly used fitting models (Weibull and Korsmayer–Peppas model). Since the calculation of the first derivatives indicated a multi-region release, we believe that the overall release mechanism is complex and is governed by several physical phenomena (at least diffusion of the respective active ingredients, the swelling and potential partial degradation of the electrospun sample). Therefore, we put this calculation into the Appendix A.

Combining now the results of the performed antimicrobial, antioxidant and release testing, it is clear that the amount of the incorporated chitosan with the trapped CAT and RES, was sufficiently effective to present a promising formulation for further consideration towards a potential clinical application in wound care. The latter is most important, considering the relatively small amounts of both active substances being released and already active at this concentration. Further studies will be also necessary in order to fully understand the evolution of the antioxidative effect over time, and with the changes in the release performance. It is expected that the initial fast release is enough to provide a beneficial decrease in the oxidative stress in the wound (a highly efficient antioxidative effect was shown for at least 60 min), whereas this effect is then sustained for the duration of 24 h, even with a potentially diminished antioxidative effect (e.g., a 9% drop in the latter was shown in the case of the formulation with incorporated RES).

## 4. Conclusions

Using the ionic gelation, the chitosan particles (CSNP), chitosan particles with embedded catechin (CSNP_CAT) and with included resveratrol (CSNP_RES), were mixed with polyethylene oxide, and used in an electrospinning device in order to manufacture the nanofibres/nanocoatings deposited on the polypropylene and viscose matrix.

SEM figures show uniform nanofilm on the surface, thus indicating the successful incorporation of chitosan particles into the formed fibre-layer. ATR-FTIR spectra of functionalised electrospun samples mostly confirmed the presence of characteristic functional groups. The XPS results showed the incorporation of nitrogen and oxygen, observed as an atomic % increase on the surface, indicating the presence of chitosan and (poly)-phenols in the formed fibre-composite material. Electrospun samples with incorporated (poly)-phenolic component (CSNP_CAT:PEO = 2:1, CSNP_RES:PEO = 2:1) indicated 90% of antioxidant activity, compared to samples without active substances, showing inhibition on ABTS radicals below 40%. Prepared nano (coated)-functionalised composites (chitosan particles with/without (poly)-phenolic substances) exhibited antimicrobial properties, since the samples, within the first hour of exposure, inhibited the growth of both used microorganisms, *Escherichia coli* and *Staphylococcus aureus*, by 99% and 83%, respectively.

The majority of the entrapped (poly)-phenolic substances were released from the electrospun samples within time intervals, which are inside the time frames for common wound dressing change frequency. In addition, the in vitro study indicates that the formulations led to a complex release mechanism, which might be further beneficial for treatment of infected and inflamed wounds, since a fast initial release provides a rapid therapeutic effect onset, which is then maintained for up to 24 h. The produced nanofibre composites exhibited a synergistic effect, since the samples showed excellent antimicrobial properties on one hand and antioxidant efficiency on the other hand. In addition to this, a controlled release of the active substances (most likely partially from the particle interior and partially from the composite surfaces) was enabled. The developed nanofibrous composites, composed of chitosan and (poly)-phenolic substances such as catechin or resveratrol, show high potential in the development of innovative textiles used in medical applications, specifically in the field of healing and treatment of skin wounds.

## Figures and Tables

**Figure 1 materials-13-02631-f001:**
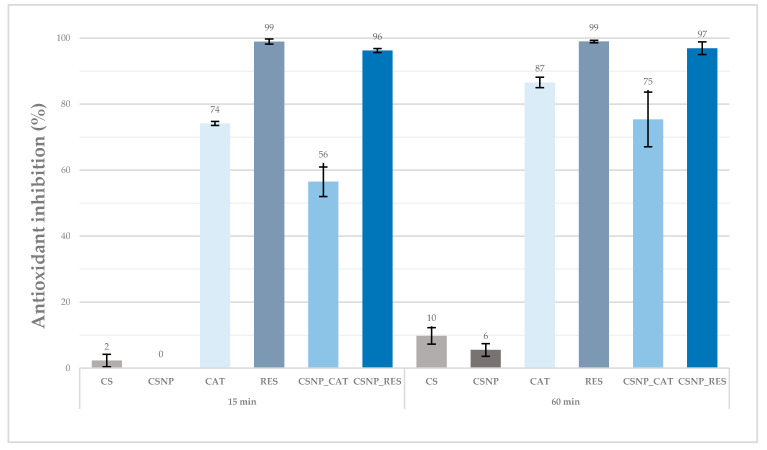
Antioxidant effect (as % antioxidant inhibition) of chitosan solution (CS; c = 10 g/L), dispersion of chitosan particles (CSNP; c = 5 g/L), chitosan particles with catechin (CSNP_CAT; c = 10 g/L) and chitosan nanoparticles with resveratrol (CSNP_RES; c = 20 g/L) after 15 and 60 min. Columns display mean value, bars represent standard deviation.

**Figure 2 materials-13-02631-f002:**
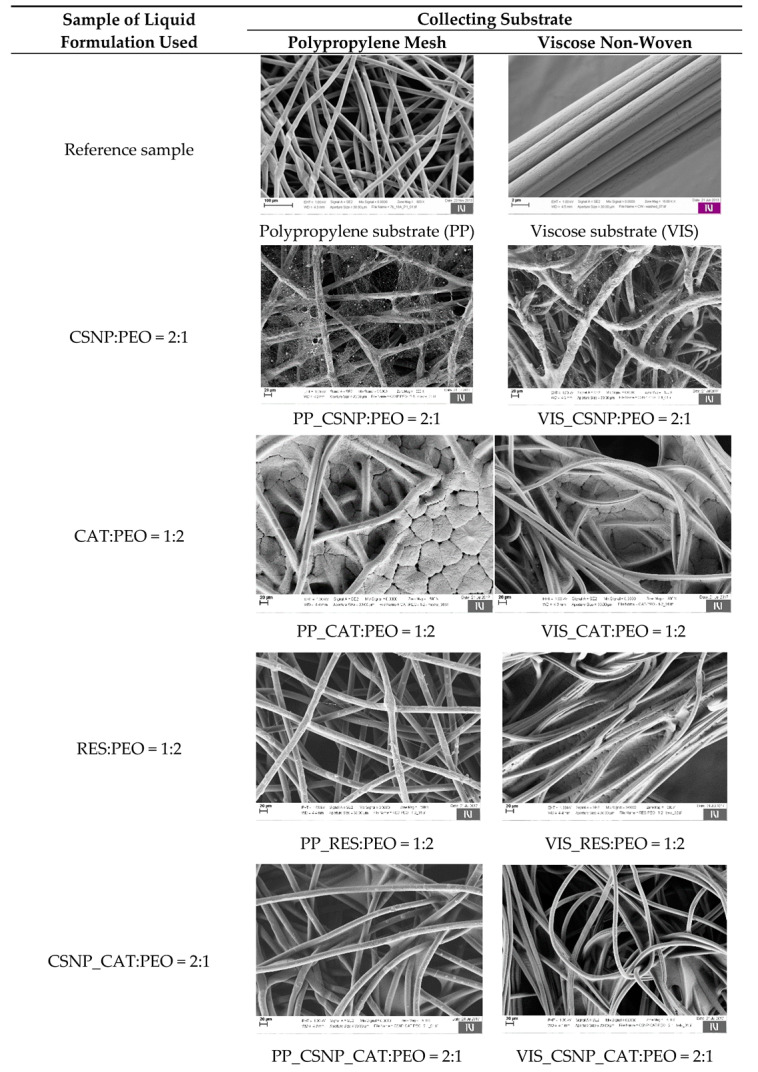
Scanning electron microscopy (SEM) images of liquid formulation samples CSNP:PEO = 2:1, CAT:PEO = 1:2, RES:PEO = 1:2, CSNP_CAT:PEO = 2:1 and CSNP_RES:PEO = 2:1, electrospun onto the polypropylene substrate material (left) and on viscose substrate (right).

**Figure 3 materials-13-02631-f003:**
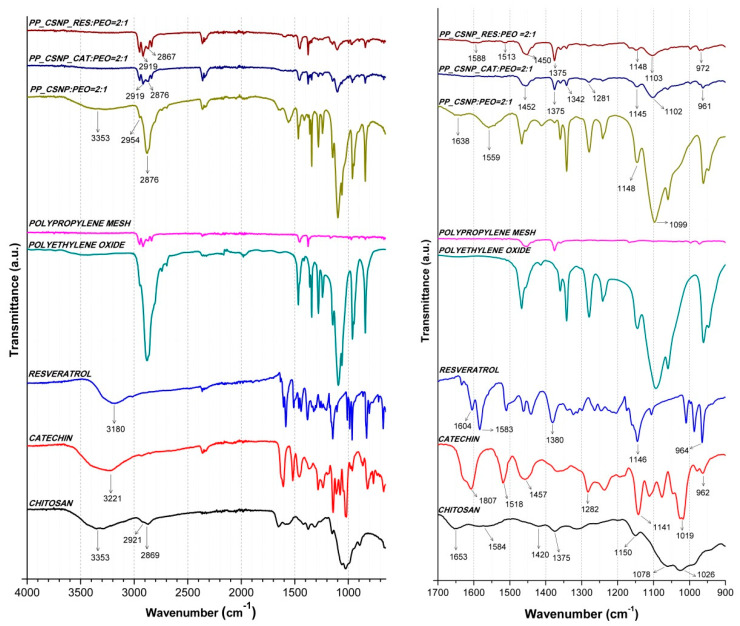
The attenuated total reflectance Fourier transform infrared spectroscopy (ATR-FTIR) spectra of chitosan (powder), catechin (powder), resveratrol (powder), polyethylene oxide (PEO; powder), polypropylene substrate and PP_CSNP:PEO = 2:1, PP_CSNP_CAT:PEO = 2:1 and PP_CSNP_RES:PEO = 2:1 electrospun samples. On the left the spectra in range from 4000 cm^−1^ up to 1000 cm^−1^ are shown; on the right the spectra in range between 1700 cm^−1^ and 900 cm^−1^ are shown.

**Figure 4 materials-13-02631-f004:**
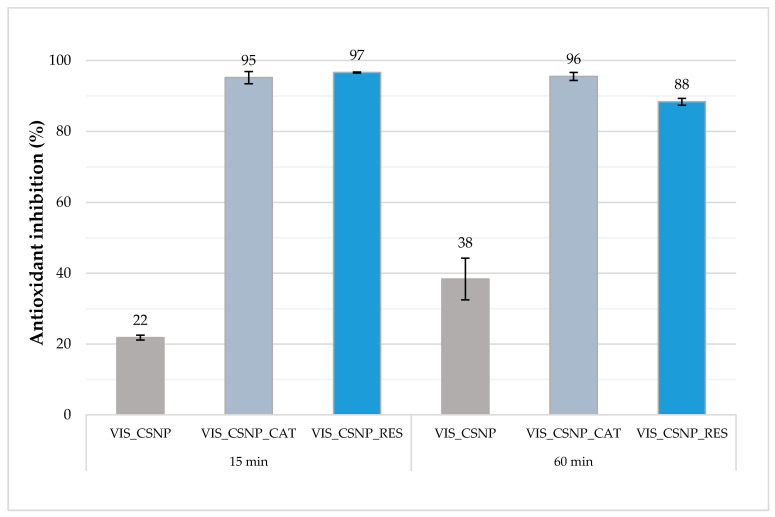
Antioxidant effect (as % antioxidant inhibition) of prepared electrospun samples of VIS_CSNP:PEO = 2:1, VIS_CSNP_CAT:PEO = 2:1 and VIS_CSNP_RES:PEO = 2:1, after 15 and 60 min. Columns display mean value, bars represent standard deviation.

**Figure 5 materials-13-02631-f005:**
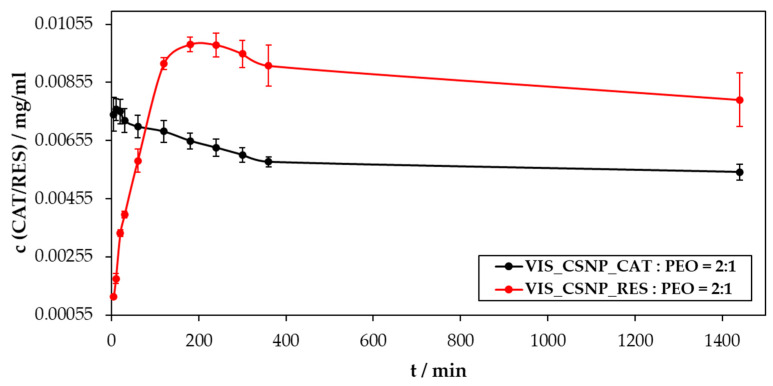
The concentration of the released substances (catechin (CAT) and resveratrol (RES)) from VIS_CSNP_CAT:PEO = 2:1 and VIS_CSNP_RES:PEO = 2:1 samples, as a function of time.

**Figure 6 materials-13-02631-f006:**
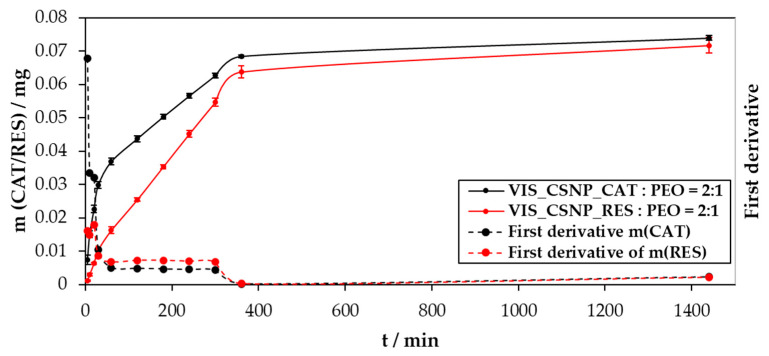
The cumulative mass and the calculated first derivatives of the obtained released substances (CAT and RES) from VIS_CSNP_CAT:PEO = 2:1 and VIS_CSNP_RES:PEO = 2:1 samples, as a function of time.

**Figure 7 materials-13-02631-f007:**
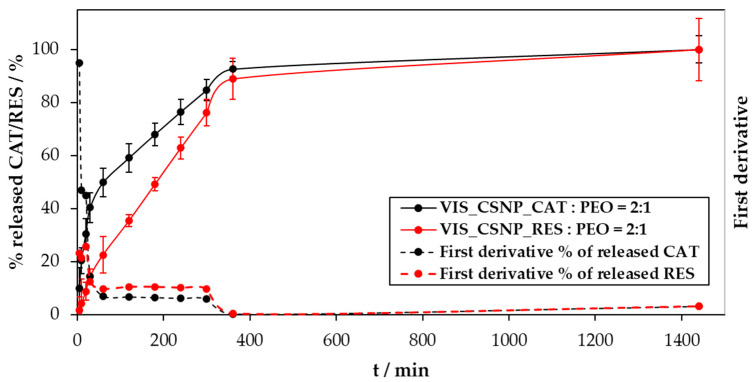
The percentage of the released substances (catechin and resveratrol) from VIS_CSNP_CAT:PEO = 2:1 and VIS_CSNP_RES:PEO = 2:1 samples, as a function of time.

**Table 1 materials-13-02631-t001:** List of electrospun fibrous sample notation and description.

Sample Notation	Description of Sample
PP_CSNP:PEO = 2:1	Polypropylene mesh coated with a dispersion of chitosan nanoparticles and polyethylene oxide (dispersion prepared in volume ratio CSNP:PEO = 2:1 (*v*/*v*))
PP_CAT:PEO = 1:2	Polypropylene mesh coated with a catechin and polyethylene oxide dispersion (prepared in volume ratio CAT:PEO = 1:2 (*v*/*v*))
PP_RES:PEO = 1:2	Polypropylene mesh coated with a resveratrol and polyethylene oxide dispersion (prepared in volume ratio RES:PEO = 1:2 (*v*/*v*))
PP_CSNP_CAT:PEO = 2:1	Polypropylene mesh coated with a dispersion of chitosan nanoparticles with embedded catechin and polyethylene oxide (prepared in volume ratio CSNP_CAT:PEO = 2:1 (*v*/*v*))
PP_CSNP_RES:PEO = 2:1	Polypropylene mesh coated with a dispersion of chitosan nanoparticles with embedded resveratrol and polyethylene oxide (prepared in volume ratio CSNP_RES:PEO = 2:1 (*v*/*v*))
VIS_CSNP:PEO = 2:1	Viscose non-nonwoven coated with a dispersion of chitosan nanoparticles and polyethylene oxide (dispersion prepared in volume ratio CSNP:PEO = 2:1 (*v*/*v*))
VIS_CSNP_CAT:PEO = 2:1	Viscose non-nonwoven coated with a dispersion of chitosan nanoparticles with embedded catechin and polyethylene oxide (prepared in volume ratio CSNP_CAT:PEO = 2:1 (*v*/*v*))
VIS_CSNP_RES:PEO = 2:1	Viscose non-nonwoven coated with a dispersion of chitosan nanoparticles with embedded resveratrol and polyethylene oxide (prepared in volume ratio CSNP_RES:PEO = 2:1 (*v*/*v*))

**Table 2 materials-13-02631-t002:** Particle size, zeta potential (ZP) and polydispersity index (PDI) of dispersions: chitosan nanoparticles (CSNP), chitosan nanoparticles with incorporated catechin (CSNP_CAT) and chitosan nanoparticles with incorporated resveratrol (CSNP_RES).

Sample	$\bar{d_{H}}$	ZP	PDI
	(nm)	(mV)	
CSNP	379.7 ± 37.1	32.4 ± 1.3	1 ± 0.0
CSNP_CAT	2986.7 ± 1139.3	11.4 ± 0.3	0.5 ± 0.1
CSNP_RES	1874.5 ± 61.5	42.2 ± 1.6	0.8 ± 0.0

**Table 3 materials-13-02631-t003:** Encapsulation efficiency (EE) of catechin (CAT) and resveratrol (RES) entrapped in chitosan particles.

Active Substance	EE (%)
CAT	10.8 ± 0.9
RES	62.7 ± 1.1

**Table 4 materials-13-02631-t004:** Minimal inhibitory concentration (MIC) of chitosan (CS), chitosan nanoparticles (CSNP), catechin (CAT) and resveratrol (RES).

Microorganism	Minimal Inhibitory Concentration (MIC) (mg/mL)			
	CS	CSNP	CAT	RES
*Staphylococcus aureus*	0.0053 ± 0.0011	0.0092 ± 0.0029	2.5 ± 0.075	0.16 ± 0.064
*Escherichia coli*	0.0039 ± 0.0001	0.0078 ± 0.0002	10.0 ± 0.5	5.0 ± 0.1

**Table 5 materials-13-02631-t005:** Conductivity (σ), viscosity (η) and surface tension (γ) of liquid formulations, namely chitosan particles (CSNP), catechin solution (CAT), chitosan particles with embedded catechin (CSNP_CAT), resveratrol solution (RES) and chitosan particles with entrapped resveratrol (CSNP_RES) and polyethylene oxide (PEO), depending on volume ratio.

Sample	Physical Properties of Liquid Formulations		
	σ (μS/cm)	η (mPa s)	γ (mN/m)
CSNP	486.0 ± 2.0	31.2	54.8 ± 0.3
CSNP:PEO = 1:1	305.0 ± 2.0	1579.8	60.5 ± 1.4
CSNP:PEO = 2:1	341.7 ± 0.6	693.0	60.9 ± 0.8
CSNP:PEO = 5:1	343.0 ± 2.0	448.7	56.1 ± 1.2
CAT:PEO = 1:2	13.2 ± 0.9	5582.3 ± 4676.9	35.8 ± 0.8
RES:PEO = 1:2	12.9 ± 1.8	6777.3 ± 301.6	35.3 ± 0.6
CSNP_CAT:PEO = 2:1	134.1 ± 0.6	679.2 ± 407.3	40.3 ± 0.2
CSNP_RES:PEO = 2:1	136.5 ± 0.8	583.5 ± 97.0	41.0 ± 0.4

**Table 6 materials-13-02631-t006:** Elemental surface composition of the formed nanofibres on polypropylene substrate (in atomic %).

Sample	Elementary Surface Composition (Atomic %) *						
	C	O	Na	Si	P	Ca	N
Polypropylene substrate (PP)	67.8	21.6	-	4.2	6.4	-	-
PP_CSNP:PEO = 2:1	63.9	28.8	1.0	0.6	1.1	0.3	3.1
PP_CAT:PEO = 1:2	76.8	18.1	0.2	2.4	1.3	1.3	-
PP_RES:PEO = 1:2	68.1	22.5	0.2	3.9	5.1	0.3	-
PP_CSNP_CAT:PEO = 2:1	80.9	16.5	0.6	0.6	-	-	1.5
PP_CSNP_RES:PEO = 2:1	70.7	23.2	1.4	0.8	1.5	0.4	1.9

* The standard deviation was within the range of 1–3%.

**Table 7 materials-13-02631-t007:** Antimicrobial activity (expressed as reduction R, %) of liquid formulation CSNP:PEO = 2:1, CAT:PEO = 1:2, RES:PEO = 1:2, CSNP_CAT:PEO = 2:1, CSNP_RES:PEO = 2:1, electrospun onto viscose substrate, against *Escherichia coli* and *Staphylococcus aureus,* as a function of time.

Sample	Reduction R (%) for Bacterial Cultures		
	*Escherichia Coli*		*Staphylococcus Aureus*
	1 h	6 h	1 h
VIS_CSNP:PEO = 2:1	>99.9	>99.9	84.8
VIS_CAT:PEO = 1:2	4.2	22.4	−17.20
VIS_RES:PEO = 1:2	22.1	40	−9.54
VIS_CSNP_CAT:PEO = 2:1	99.3	>99.9	84.3
VIS_CSNP_RES:PEO = 2:1	>99.9	>99.9	83.9

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
