# Peer review of "Electrospun Composite Nanofibrous Materials Based on (Poly)-Phenol-Polysaccharide Formulations for Potential Wound Treatment"

_materials, 2020, doi:10.3390/ma13112631_

Round 1

Reviewer 1 Report

After analysis of the manuscript entitled: "Electrospun composite nanofibrous materials based on (poly) - phenol - polysaccharide formulations for potential wound treatment" I identify the follow major issues: The manuscript is too extensive in general, not focus on the main subject (the production of Electrospun composite nanofibrous materials based on (poly) - phenol - polysaccharide formulations) which leads to the ideas not being presented clearly. I could choose some examples like:  - The introduction section has several pages when 1 full-page could be considered too much; - FTIR analysis is not focused on the main goal of the research work and it is too descriptive; - The conclusion must be short and concise.

I reject the paper because I believe that the article is not ready to be published being needed changes that will lead to a massive alteration of the manuscript.

Reviewer 2 Report

The work has been prepared correctly, but requires several minor fixes:

  • Introduction and Conclusion sections should be shortened.
  • Table 1 should be deleted due to the duplication of information, and shortcuts should appear in the place where the term first appears.
  • The molecular weight of chitosan should be given.
  • The sentence: "To dissolve chitosan better, the pH of the solution was adjusted to 3.8 with the addition of concentrated acetic acid" should be changed for: "To dissolve chitosan, the pH of the solution was adjusted to 3.8 with the addition of concentrated acetic acid".

Reviewer 3 Report

Very interesting, complete work showing the preparation of electrospun composite nanofibrous materials based on (poly) – phenol - polysaccharide formulation. It has been shown that prepared composite nanofibers are ideally suited as a controlled drug delivery system, especially for local treatment of different wounds, owing to their high surface and volume porosity and small fiber diameter. As an active antioxidants were used  catechin and resveratrol were used. Both antioxidants  were inserted into chitosan particles, and further subjected to electrospinning.

Electrospun materials containing (poly) - phenolic component and indicated a high degree of antioxidant activity, since the inhibition of ABTS radicals was 90%, while electrospun material containing only chitosan particles showed inhibition on ABTS radicals below 40% (lack of any antioxidant activity). Additionally prepared functionalized materials exhibited antimicrobial activity.

The prepared nanomaterials exhibited a synergistic effect, since the samples showed excellent antimicrobial properties on one hand and antioxidant efficiency on the other hand.

Missing elements of the manuscript that should be completed:

  • The only missing element concerns information on the origin of the used bacteria strains.
  • There is also a lack of information on statistical analysis of the in vitro results.

Reviewer 4 Report

The manuscript titled: “Electrospun composite nanofibrous materials based on (poly) – phenol – polysaccharide formulations for potential wound treatment” written by Zemljic and co-workers deals with the preparation of chitosan-based nanofibrous materials produced via electrospinnning, and containing catechin and resveratrol as antioxidants. Formulations were characterized and the bioactive properties evaluated in terms of antioxidative and antimicrobial properties thought in vitro tests. Results are promising in terms of wound care applications, however MAJOR revision is mandatory. In details: 1) Antioxidant effect and Figure 1. The comparison seems not effective due to the variability of the concentration of each condition examined. Additionally, at lines 462-464, and 468-475, Authors reported that there is a concentration effect in the final response. Since these data are interesting for the discussion, this Reviewer asked to completely re-organize the paragraph reporting the effect of composition vs. concentration vs. time. Data should be compared by fixing each parameter otherwise it is impossible to rationalize any behavior (e.g., in Figure 1 concentrations are random and this way it is difficult to compare). 2) MIC and Table 5. Also the minimal inhibitory concentration of chitosan should be provided. 3) Figure 2. Micrographs should be collected at the same magnification and marker should be readable. Additionally, micrographs of bare reference substrates (PP and VIS) should be included. 4) Rather than FTIR is better to consider it as ATR. 5) Figure 3. Spectra are impossible to read, thus figure should be changed. A solution could be to cut the spectra and highlight specific section and peaks in order to follow the growth/shift of signals. Furthermore, basing on the data in Figure 3, it is not possible to confirm the presence of CAT in PP_CSNP_CAT:PEO sample. 6) It is hard to believe that the spectrum of PP_CSNP:PEO did not shown the signal of PP. Moreover, the signals of PP_CSNP:PEO are different with bare chitosan, why? Is it possible to have also a spectrum of PEO? 7) Authors should provide the same FTIR analysis even for samples on VIS. 8) The utility of paragraph 3.2.3 is negligible. Additionally, authors cannot state at lines 689-691: “The results pointed out that (poly) – phenols, during their incorporation into chitosan particles, reacted with chitosan and even, most probably, had covered the chitosan particles” as the formation of new bonds should be detected by other techniques. Please clarify this point. 9) Figure 4. Are data obtained at the same concentration (see point 1)? Is it possible to have also the data for PP substrates? 10) Paragraph 3.2.5. Also the reduction (%) of S. aureus at 6h should be provided. 11) Figure 5. It is hard to believe that the correlation has a meaning (even in the case of S. aureus). Probably more points are necessary.

Round 2

Reviewer 1 Report

There was an extensive improvement on the manuscript however there are yet some issues that need to be addressed namely:

-Figure 1- what are the lines in the bars? Standard deviations? How did you do it? It is supposed to use software to do it and not do it by yourself (why are the lines not aligned?)

-Please provide statistical analysis every time that is necessary and enrich your manuscript with this information;

-All your FT-IR analyses do not present any Reference. Please fundament your discussion with references;

-Topic 3.2.4 and 3.2.5 – Please discuss your results. Why do you have that results? Ideally, it was interesting if you can base your discussion with literature. I remind you that the topic is: “Results and discussion”.

Reviewer 4 Report

The revised version of the document "Electrospun composite nanofibrous materials based on (poly) - phenol - polysaccharide formulations for potential wound treatment" has been submitted for revision. The quality is significantly improved, however still MAJOR revision is mandatory. In detail:

1) Figure 2. SEM analysis. The micrographs of all samples must present the same magnification (even the references). Additionally, markers are still not readable, try to modify all figures in order to allow readers to see the marker.

2) Figure 3. ATR-FTIR. The second half of the image is not properly cut as the x-axis unit is not readable completely. Please modify the figure in a manner suitable for publication (try to better the quality).

3) In the past revision, this Reviewer asked for the addition of reduction data against S. aureus after 6 h (analogously as for E. coli). In this revision, authors reply that 6 h tests are not necessary. Therefore, this reviewer have to stess again that Authors compare two different types of microorgamisms, with different Gram-nature and characteristics. If 6 h data are not necessary, why presenting the data for E. coli? There are two options: either Authors specify the reason why tests at 6 hours are not necessary and consequently removed the data also for E. coli, or they should add the data at 6 hours also for S. aureus.

4) Figure 5. The trends in the figure have no meaning as the measurements require more data. In fact, the reason why the Coefficient is close to or far from 1 according to the tests performed by Authors depends only on ONE single point, and this is not enough!!!! Even here, there are two options: either the Authors added more points to demonstrate the behavior, or they should remove the entire figure (and relative data discussion).

Round 3

Reviewer 1 Report

Regarding the statistical analysis could the authors be more precise about the tests that they run. Where do they apply it?

Please add this information on the methods section as well as across the manuscript. Also on the methods section describe the software that you use it.

Reviewer 4 Report

The revised version of the manuscript is significantly improved thus it can be accepted for publication. 

Author Response

Thank you very much for your approval and all the previous suggestions that help us to improve the article.